# A Six-Year Analysis of Biological Therapy for Severe Psoriasis in a Lithuanian Reference Centre of Dermatovenereology

**DOI:** 10.3390/medicina56060275

**Published:** 2020-06-04

**Authors:** Tadas Raudonis, Akvile Gliebute, Anna Greta Grigaityte, Zivile Lukosiunaite, Tatjana Karmaziene, Jurate Grigaitiene

**Affiliations:** Centre of Dermatovenereology, Vilnius University, LT-08411 Vilnius, Lithuania; tadas.raudonis@santa.lt (T.R.); anna.grigaityte@gmail.com (A.G.G.); zivile.lukosiunaite@gmail.com (Z.L.); tatjana.orlovskyte@gmail.com (T.K.); jurate.grigaitiene@santa.lt (J.G.)

**Keywords:** severe psoriasis, PASI, biological therapy, efficacy, safety

## Abstract

*Background and Objectives*: Biological therapy is widely used for the treatment of severe psoriasis. The objective of this study was to evaluate the efficacy and safety of biological therapy for patients with severe psoriasis. *Materials and Methods*: A retrospective study of 79 patients with severe psoriasis, who have been treated with biological therapy between 2012 and 2018, was conducted. During this study, the following data were collected and evaluated: sex, age, body mass index (BMI), duration of illness, the results of treatment with biological therapy, concomitant therapy, Psoriasis Area and Severity Index (PASI) and adverse events. *Results*: In total, 74.7% (*n* = 59) of subjects were male. Their overall average age was 47.4 ± 11.4 (range: 18–73) years. Their baseline BMI was 27.6 ± 5.9, which increased to 29.6 ± 4.5 after 6 years of treatment. The mean duration of psoriasis was 25.7 ± 12.5 years. In total, 39.2% (*n* = 31) of subjects received infliximab, 36.7% (*n* = 29)—etanercept, 24.1% (*n* = 19)—ustekinumab. The treatment duration for infliximab, etanercept and ustekinumab was 201.6 ± 86.8, 156.2 ± 137.4 and 219.1 ± 95.7 weeks (*p* < 0.01), respectively. Overall, 65.8% (*n* = 52) of subjects were also on methotrexate; 30.8% (*n* = 16) of them discontinued it due to clinical improvement (31.3% (*n* = 5)), impaired liver function (31.3% (*n* = 5)), and intolerance (25% (*n* = 4)). Baseline PASI was 20.8 ± 8.8. PASI 50 was achieved by 96.2% (*n* = 76) of patients at week 11, PASI 75 by 86.1% (*n* = 68) at week 16, PASI 90 by 54.4% (*n* = 43) at week 35, and PASI 100 by 13.9% (*n* = 11) at week 33. The overall incidence rate of adverse events was 0.362 per patient year of follow-up. *Conclusion*: Biological therapy is an effective and safe treatment for patients with severe psoriasis.

## 1. Introduction

Psoriasis is a chronic inflammatory and autoimmune disease that causes thickened, scaly plaques on the skin. It generally affects about 1–3% of the population worldwide [1,2]. Although, in most cases, the disease manifests on the skin, it is also associated with other significant comorbidities, including psoriatic arthritis, cardiovascular disease, inflammatory bowel disease, diabetes, depression, and cancer [2,3,4,5]. The Psoriasis Area and Severity Index (PASI) is used for the measurement of psoriasis severity. A PASI score below 10 defines psoriasis as mild, 10 to 15 as moderate and above 15 as severe [6]. There are three main groups of treatment options: topical treatment, phototherapy and systemic treatment. While mild forms can be managed by topical therapy, moderate to severe psoriasis typically requires phototherapy and (or) systemic therapy. The limitations of traditional systemic medications led to the development of alternative treatment options due to its often-intolerable adverse effects and organ toxicity (5–50% of cases). Moreover, traditional treatment might be ineffective [7,8,9]. For patients with moderate to severe psoriasis, biological therapy is prescribed when they do not tolerate the traditional systemic treatment because of its adverse effects or inefficacy [10]. Biologics are monoclonal antibodies targeting cytokines, such as interleukin (IL)-17, IL-23 and tumor necrosis factor alpha (TNF-α) [7]. Due to its great efficacy and tolerance, biological therapy is increasingly used for moderate to severe psoriasis [11]. The aim of this study is to evaluate the demographic data of patients with severe psoriasis who were treated with biological therapy, including prescribed biologics, baseline PASI, dermatological quality of life index (DLQI), body mass index (BMI), laboratory and instrumental examination results and their changes during the therapy, adverse effects, comorbidities and their treatment.

## 2. Materials and Methods

A retrospective and prospective medical record analysis was conducted on 79 patients who have been treated with biological therapy between 2012 and 2018 at Vilnius University Hospital Santaros Klinikos, Centre of Dermatovenereology (VUH SK DVC). The study was approved by the Regional Biomedical Research Ethics Committee (approval number 158200-17-974-471). In total, 79 patients aged between 18 and 70 years were included. All patients were diagnosed with severe psoriasis (PASI higher than 15) and have been treated with biological therapy.

The following data were recorded: sex, age, height, weight, BMI, illness and medical history, PASI, laboratory and instrumental test results, current treatment of psoriasis and its adverse effects. Statistical analysis was conducted with Microsoft Excel (2004, Redmond, WA, USA) and SPSS 23.0 (IBM Corporation, Armonk, New York, NY, USA). The statistically significant value was *p* < 0.05.

## 3. Results

### 3.1. Baseline Demographic and Other Clinical Characteristics

Out of 79 subjects, 74.7% (*n* = 59) were men. The average age of subjects was 47.4 ± 11.4 (age range 26–73 years). In total, 32.9% of patients had a positive family history of psoriasis and 12.7% (*n* = 12) were smokers. The total mean duration of morbidity was 25.7 ± 12.5 years, and the longest mean duration of psoriasis morbidity was among patients treated with ustekinumab—3.5 ± 13.9 years. The most common sites of lesions were arms (96.2%), legs (94.9%), torso (88.6%) and scalp (87.3%). Psoriatic onychodystrophy were recorded in 81% (*n* = 64, *p* = 0.555) of patients and psoriatic arthropathy in 72.2% (*n* = 57, *p* = 0.226) of patients. Our patients were treated with two groups of biologics: TNF-α (etanercept, infliximab) and IL-12/23 (ustekinumab) inhibitors. Overall, 39.2% (*n* = 31) patients were treated with infliximab, 36.7%. (*n* = 29)–etanercept and 24.1% (*n* = 19)–ustekinumab. The mean duration of biological therapy was 143.7 ± 112.4 weeks and the longest used biologic was ustekinumab—219.1 ± 95.7 weeks. In 6 years, 15.2% (*n* = 12) of patients were switched to another biological agent (mainly to ustekinumab (41.7%, *n* = 5)) and, for a second time, the treatment was changed for another two patients (both to ustekinumab). All baseline demographic and clinical characteristics of patients are presented in Table 1.

### 3.2. Baseline PASI and Changes during Treatment

The overall mean PASI was 20.8 ± 8.8 at the initiation of biological therapy, but it differed among different biological agents (*p* = 0.025): ustekinumab—19.2 (±8.9), etanercept—24.3 (±9.3); infliximab—18.5 (±7.5). After one year of treatment, the average PASI was 5.25 (*n* = 56, *p* = 0.178) and it remained quite stable throughout the 6 years of follow-up (Figure 1).

### 3.3. Efficacy of Different Biological Agents

In the etanercept group, the highest proportion of patients reached PASI 50 (59%) and PASI 75 (38%) after four months of treatment, while 14% of subjects reached PASI 90 after one year of treatment, which was the highest number (Figure 2).

During the first year of treatment with ustekinumab, PASI 50, PASI 75 and PASI 90 were achieved in the majority of patients (84%, 53%, 26%, respectively). In this group, the highest number of patients reached PASI 50 (84%) and PASI 75 (47%) compared to other biological agents (Figure 2).

The majority of patients in the infliximab group reached PASI 50 (56%) and PASI 75 (44%) after two years of treatment, whereas the best treatment results in this group were during the first year of treatment—36% achieved PASI 90. In this group, the highest number of patients reached PASI 90 compared to other biological agents (Figure 2).

### 3.4. Baseline DLQI and Its Changes during Treatment

From 2018, DLQI evaluation became mandatory in Lithuania. At baseline, mean DLQI was 14.7 ± 7.3 (*n* = 25). Changes in DLQI averages were calculated after 1, 3, 6 and 9 months of treatment. The total change in mean DLQI after one month was −7.26 and, after 9 months, it increased to −10.6. Infliximab (*n* = 11 → 7) showed the largest improvement in scores, and the lowest impact on DLQI was observed in the ustekinumab (*n* = 1) group. All DLQI mean changes are presented in Figure 3.

### 3.5. Baseline BMI and Changes over the Course of Biological Therapy

Mean body mass index (BMI) was 27.6 ± 6 kg/m^2^ at baseline. After one year of treatment, the average BMI was 28.0 ± 4.1 and, after six years, it was 29.6 ± 4.5 (*p* = 0.039).

### 3.6. Laboratory and Instrumental Tests

#### Changes in Complete Blood Count

Changes in the following parameters were evaluated: complete blood count, erythrocyte sedimentation rate (ESR), C-reactive protein (CRP), glucose, cholesterol and liver function tests. The changes in mean counts were insignificant. The laboratory parameters were monitored at each visit (every 1–3 months); the mean changes were evaluated annually. The means of all results are shown in Table 2.

Patients were screened for latent and active tuberculosis infection annually with a chest X-ray, tuberculin skin test (TST) and/or gamma interferon blood test (QuantiFERON- tuberculosis (TB) Gold test). None of the patients showed tuberculosis-specific findings in chest X-rays. A positive TST was detected in nine patients and six patients tested positive for QuantiFERON-TB Gold. After the positive test results, patients were referred to a pulmonologist for further workup and treatment. A false positive TST was determined in five patients.

An abdominal ultrasound was performed annually during treatment and new changes were recorded in 21.6% (*n* = 17) of patients. The most common finding was hepatic steatosis—12.7% of cases (*n* = 10) and pancreatic steatosis—12.7% of cases (*n* = 10). Other more common pathological changes were as follows: hepatomegaly—8.9% (*n* = 7), kidney cysts—6.35% (*n* = 5), splenomegaly—5.1% (*n* = 4), gall bladder polyps—3.8% (*n* = 3) and pancreatic stiffness in 3.8% of cases (*n* = 3).

### 3.7. Adverse Effects of Biological Therapy

The incidence of adverse effects was 0.362 cases per patient year of follow-up. The majority of adverse reactions were reported in patients treated with etanercept (53.8%, *n* = 34), and the lowest number occurred in the infliximab group (22.5%, *n* = 10) (Table 3).

The most common adverse effects caused by biological therapy were infections of the upper respiratory tract—0.334 cases per patient year. In the etanercept group, the incidence was 0.436 per patient year, ustekinumab—0.337, infliximab—0.151.

Five patients were diagnosed with latent tuberculosis infection (LTBI). Two patients were treated with etanercept (Enbrel^®^) and had a positive tuberculin skin test (TST) and QuantiFeron-TB (QFT) results after one year, and after 5 years of treatment with biological therapy. The three patients being treated with ustekinumab (Stelara^®^) had positive tuberculosis tests: one patient had a positive TST result after 2 years of treatment and another 2 had positive TST and QFT test results after 5 years of treatment. All of them had no findings suggestive of TB in their chest X–rays. All patients who were diagnosed with LTBI received 3-month prophylaxis with isoniazid (150 mg 3 times daily) and rifampicin (300 mg twice daily), during which biological therapy was stopped.

### 3.8. Switching of Biologics

Twelve patients required switching to another biological agent due to insufficient efficacy. As a first-line therapy, 75% (*n* = 9) of them received etanercept and 25% (*n* = 3) infliximab. In total, 41.5% (*n* = 5) of patients were switched from etanercept to ustekinumab, 33.2% (*n* = 4) from etanercept to infliximab and 24.9% (*n* = 3) from infliximab to etanercept. Prior to administration of the second biological agent, the mean PASI for all patients was 16.9 ± 6.1. After the treatment was changed, PASI 75 was achieved in 72.7% (*n* = 8) of patients treated with a second biological agent in 19.9 ± 12.4 weeks. Of these, 80% (*n* = 4) were treated with ustekinumab, 66.7% (*n* = 2) etanercept and 66.7% (*n* = 2)–infliximab. Switching to a third biological agent was indicated in two patients treated with infliximab and etanercept, both were assigned to ustekinumab. Before treatment with the third biological agent, the PASI average was 19.8. One patient achieved PASI 50 in 12 weeks and PASI 75 in 24 weeks. The other patient did not respond clinically with his PASI, changing from 15 at baseline to 15.3 after 24 weeks.

### 3.9. Comorbidities

In total, 31.6% (*n* = 25, *p* = 0.062) of patients had dermatological comorbidities, the most common being viral warts (Verruca vulgaris)—10.1% (*n* = 8) and seborrheic dermatitis—also 10.1% (*n* = 8) of cases. Overall, 63.3% (*n* = 50, *p* = 0.867) of patients had non-dermatological comorbidities, while 40.5% (*n* = 32) of them had primary arterial hypertension. All concomitant diseases are presented in Table 4.

### 3.10. Additional Topical Treatment and Phototherapy

Topical treatment was prescribed for all patients during treatment with biological therapy. Almost all patients were treated with emollients (98.7%), and 84.8% (*n* = 67) were treated with topical glucocorticoids. More than half of patients used keratolytic agents and only 5.1% (*n* = 4) of the subjects used calcineurin inhibitors. UVB 311 nm phototherapy was prescribed for 46.8% (*n* = 37), of which 45.6% was whole body and 26.6% (*n* = 21) was local phototherapy. All data on additional topical treatments and phototherapy are presented in Table 5.

### 3.11. The Use of Methotrexate before and during Biological Therapy

Overall, 74.7% (*n* = 59, *p* = 0.003) of patients were on methotrexate prior to biological therapy. Most of them (84.7% (*n* = 50)) were taking it orally. The average treatment duration was 114.7 (±138.7) weeks, with an average dose of 15.2 mg (range from 5 mg to 30 mg). Prior to biological therapy, the dose of methotrexate was not adjusted for 59.3% (*n* = 35) of the patients, and decreased in 6.9% (*n* = 10) or increased in 23.7% (*n* = 14) (to the maximum dose of 30 mg) of the subjects. The most common identified adverse effects of methotrexate were nausea—3.3% (*n* = 14), increased levels of liver enzymes—15% (*n* = 9), thirst—5% (*n* = 3), weight loss, severe headache and somnolence—3.3% (*n* = 2), general weakness and leukopenia—1.7% (*n* = 1).

During the treatment with biological therapy, methotrexate was continued in 65.8% (*n* = 52) of patients, mainly in the infliximab group. The mean dose of methotrexate after initiation of the biological therapy was 12.12 (±5.1) mg (range between 5 and 25 mg), and the average duration of combined treatment with these medications was 79.9 (± 69.2) weeks. During the treatment, methotrexate was discontinued in one third (30.8%, *n* = 16) of the patients receiving it. The most commonly reported causes of this treatment termination were as follows: regression of the disease—31.3% (*n* = 5), increase in the liver enzyme levels—31.3% (*n* = 5), intolerance—25% (*n* = 4), and tuberculosis—6.3% (*n* = 1). The dose of methotrexate was decreased in 46.2% (*n* = 24) of patients during treatment, and the most common causes for it were elevated liver enzymes (37.5%, *n* = 9), regression of symptoms (33.3%, *n* = 8) and intolerance of the medication (16.7%, *n* = 4). The dose was increased during treatment for 13.5% (*n* = 7) of patients; this was due to their deterioration during treatment (71.4%, *n* = 5).

### 3.12. Consultations by other Specialists and Medications Used

The majority of patients were consulted by cardiologists—30.4% (*n* = 24) and rheumatologists—8.9% (*n* = 7). The most commonly taken medications (unrelated to psoriasis) were antihypertensives (used by 38% of subjects (*n* = 30)). In total, 30.4% (*n* = 24) of patients had used hepatoprotectors, and antifungals—17.7% (*n* = 14). All numbers of consultations of other specialists and medications used are presented in Table 6.

### 3.13. Drug Survival Rates of Biological Agents

The drug survival for all biological agents decreased over time. After the first year, drug survival of ustekinumab was 100%, which dropped to 95.4% after the second year. After three years, it decreased to 90.9% and after four years it reached 86.3%, then remained unchanged after five years. The 6-year mean annual drug survival rates of etanercept were 95.4%, 81.8%, 72.7%, 68.1%, 59% and 50%, respectively. After the first year, the drug survival of infliximab was 88.4%, which decreased to 84.6% after the second year and to 80.7% after the third year. In our study, ustekinumab was associated with the highest drug survival rate and etanercept was associated with the lowest one (Figure 4).

## 4. Discussion

Biologics have significantly improved the treatment of psoriasis. Because of its efficacy and safety, biological therapy is now the standard treatment for moderate to severe psoriasis, especially in patients who do not respond to conventional systemic treatment, such as methotrexate or retinoids. Moreover, biologics are highly effective in treating patients with nail and joint involvement [1,12]. Currently, there are no international guidelines that clearly specify first-choice biologics for moderate to severe psoriasis. As of 2017, biological therapy prescription in Lithuania is determined by the minimum annual cost of the treatment and the first line biological agents are TNF-α inhibitors [6]. However, in the same year, a French study with 830 biologically naïve patients identified patient-related factors associated with the choice of the first biological treatment. In this study, adalimumab was the first-choice treatment for patients with severe psoriasis and for those who had psoriatic arthritis. Ustekinumab use was associated with a history of tuberculosis. Young patients and patients with infectious comorbidities more frequently received ustekinumab than etanercept, while patients with cardiac comorbidities, metabolic syndrome or a history of cancer commonly received etanercept [13].

### 4.1. Efficacy

Systematic reviews from Spain in 2014 and Germany in 2015 have reported that the majority of PASI 75 response rates were seen for infliximab and ustekinumab, followed by adalimumab and etanercept [14,15]. Another review, published in the United States (US) in 2015, indicated that infliximab has the greatest PASI improvement and the fastest onset of action, followed by ustekinumab, adalimumab, and etanercept [16]. In our study, PASI 75 response rates were highest for infliximab (95.5%), followed by ustekinumab (84.2%), and etanercept (79.3%). A Cochrane metanalysis in 2017 showed that ustekinumab was superior to etanercept when reaching PASI 90. Furthermore, no clear differences were found between infliximab, adalimumab and etanercept [17], while, in our study, the highest PASI 90 response rates were for patients on infliximab (68.5%), followed by ustekinumab (52.6%) and etanercept (44.8%). These conflicting treatment efficacy results may be due to our small sample size.

### 4.2. DLQI

Patients with psoriasis are more likely to experience depression, anxiety and suicidal ideation [18]. The manifestation of these conditions is related to psoriasis severity [19]. In Lithuania, biological therapy is indicated when PASI and DLQI are equal or over 15 and 10, respectively [6]. In 2016, Australian retrospective longitudinal study results showed that improving the condition of the skin dramatically improves the quality of life of psoriasis patients. PASI and DLQI scores remained low after 12 months on biological therapy and gradually decreased further with time [18]. Another American systematic review in 2014 observed that patients who had achieved more than 75% PASI reduction showed a statistically significant mean of DLQI reduction compared with those who had achieved less than a 75% reduction in PASI [19]. Furthermore, the data of a Swiss real-life psoriasis registry study in 2016 demonstrated that, compared with standard systemic treatment, lower DLQI values were found in the patient group who were treated with biological therapy [20]. Thus, effective treatment not only improves the skin condition, but also the quality of life of the patients [18]. The results of this study also show that DLQI scores have been reduced for all patients who received biological therapy.

### 4.3. BMI

Elevated levels of cytokines caused by chronic inflammation in psoriasis also have effects on adipogenesis and lipid metabolism. Patients with moderate to severe psoriasis have a 1.8 kg/m^2^ higher BMI than individuals without psoriasis, as reported in a Norwegian study in 2018 [21]. The baseline BMI of the subjects in this study was 27.6 kg/m^2^; after 6 years of treatment it increased by 2 kg/m^2^ (up to 29.6 kg/m^2^). Nevertheless, a retrospective study of Japanese patients in 2018 showed that TNF-α inhibitors increase bodyweight, whereas IL-23 and IL-17A inhibitors do not have any impact of body weight changes during treatment with biological therapy [22]. In this study, the mean BMI value increased during treatment with both TNF-α and IL inhibitors.

### 4.4. Changes in Laboratory and Instrumental Tests

In the course of biological therapy, complete blood count changes were insignificant for all subjects. Based on a Romanian retrospective study in 2015, CRP and ESR levels significantly decreased for patients with rheumatoid arthritis during treatment with biological therapy [23]. In the results of this study, the mean CRP concentration decreased from a baseline of 7 mmol/L to 1.7 mmol/L after 6 years of treatment, whereas ESR decreased from a baseline of 12.8 mm/h to 7.5 mm/h after 3 years; however, it increased to 13.6 mm/h after 6 years of treatment. The results of a Taiwanese observation in 2015 show that biological agents have an insignificant effect on cholesterol levels [24]. However, in this study, cholesterol levels increased from baseline 5.5 mmol/L to 6.5 mmol/L after 6 years of biological therapy. The increased levels of inflammatory cytokines also have an impact on insulin secretion and resistance, though no clear correlation between psoriasis and glucose levels was found in the Norwegian study in 2018 [25,26]. The mean glucose changes in the subjects in this study were within the reference range—it started to increase slightly from 3 years; however, after 6 years of treatment, values were still within the reference interval.

According to a French review in 2017, biological agents may increase liver enzymes. However, their concentrations often vary slightly and return to normal values within continued treatment [27]. The results of this study suggest that changes in ALT, AST and ALP mean concentrations varied in reference ranges, when GGT concentration increased less than two-fold during the first 5 years of treatment, returning to the reference range of 33.4 mmol/L after 6 years of treatment. Although this study showed that biological therapy has no significant changes in laboratory tests, it is suggested that follow-up is carried out and vigilance is maintained.

Biological therapy increases the risk of active and latent tuberculosis; therefore, prior to and during the treatment, screening is mandatory. It is recommended to perform a Mantoux or QuantiFERON-TB Gold test and chest X-ray annually [28]. Comparing both Mantoux and QuantiFERON-TB Gold tests, it has been observed that the results of the Mantoux test are more likely to be false positive due to its biased evaluation in inflammatory skin diseases. In addition, immunosuppressants used in combination with biological therapy, such as methotrexate, can also lead to false negative test results, whereas QuantiFERON-TB Gold test results are not affected [29]. In this study, a positive tuberculosis test was identified 15 times, of which the false positive Mantoux test was detected five times. In the United Kingdom survey results in 2015, 10% of patients who were treated with biological therapy produced positive QuantiFERON-TB test without pathological changes in their chest X-ray [30]. As well as the results of this study, any specific radiographic changes for patients with positive tuberculosis tests were not detected.

By evaluating the ultrasound of the internal organs of all subjects annually, 21.6% of cases had new pathological abnormalities, most commonly fatty liver (12.7%, *n* = 10) and pancreas (12.7%, *n* = 10). However, no scientific articles have been published that examine new changes in the ultrasound of internal organs caused by treatment with biological therapy, meaning that we were not able to compare the results and directly relate them with the use of biological agents.

### 4.5. Safety

One of the most significant adverse events of biological therapy is tuberculosis (TB). According to a Swedish study in 2017, biological therapy increases the risk of TB fourfold [28]. It may induce the reactivation of latent TB or a de novo infection [14,28]. Clinical manifestation can be late—the averages for infliximab, adalimumab and etanercept are 3, 5 and 11.5 months, respectively [29]. It should be noted that extrapulmonary TB is more common in those who receive biological treatment; it manifests in more than 50% of all cases [28,29,30]. When different TNF-α antagonists were compared, it was reported that etanercept had the lowest incidence of TB [29]. Moreover, 2015 data from the British Society for Rheumatology Biologics Register showed that the highest rate of TB was for adalimumab (144 events/100,000 patients/year), followed by infliximab (136/100,000 patients/year) and etanercept (39/100,000 patients/year) [30]. In our study, LTBI was diagnosed in 6.3% (*n* = 5) of all patients, with a rate of 0.075 cases per patient year, most commonly treated with ustekinumab (3.8%, *n* = 3). According to the World Health Organisation (WHO) algorithm, several LTBI treatment regimens are available: isoniazid for 6 months, isoniazid and rifampicin for 3 months or rifapentine and isoniazid for 3 months. The effectiveness of all treatment options is the same [31]. In this study, all patients with LTBI (6.3%, *n* = 5) were treated with isoniazid and rifampicin for 3 months.

When determining other adverse effects of biological therapy, the results of a German systemic review in 2015 showed that the risk of serious adverse effects and infections, lymphoma and congestive heart failure in patients receiving biological therapy is not significantly different compared with the control group [14]. Other adverse effects of TNF-α inhibitors include injection site reactions, upper respiratory tract infections, drug-induced lupus, abnormal liver function tests, and palmoplantar pustulosis, which were noted in US systemic reviews in 2018 and 2019. Moreover, a minimally increased risk of non-melanoma skin cancer and other malignancies was mentioned [21,26]. In this study, the most common adverse effects of biological therapy were upper respiratory tract infections—0.334 cases per one patient/year—and basal cell carcinoma was diagnosed in one patient (1.3%). Although the US review from 2018 observed that ustekinumab may promote the occurrence of cardiovascular events, the Psoriasis Longitudinal Assessment and Registry (PSOLAR) registry from 2015 did not note this [21]. No cardiovascular or neurologic events were noted in our cohort.

### 4.6. Drug Survival

The UK study in 2015 (British Association of Dermatologists Biologic and Immunomodulators Register (BADBIR)) and US study in 2016 (PSOLAR) compared the drug survival of TNF-α and IL-12/23 inhibitors in patients with psoriasis. Both of them observed significantly higher drug survival of ustekinumab for biologically naïve and biologically experienced patients, whereas, for TNF-α inhibitors, adalimumab demonstrated the longest drug survival [32,33]. This study data indicated that the highest drug survival occurred for ustekinumab. In previously mentioned studies, predictors of treatment failure were identified, such as female gender, current smoking status, a higher baseline DLQI and being on etanercept or infliximab, while being on ustekinumab and having psoriatic arthritis led to longer drug survival. Moreover, adalimumab was compared with other TNF-α inhibitors and the results have shown that infliximab was a predictor of discontinuation due to adverse events, while etanercept caused discontinuation due to inefficiency [32,33]. The main reason for therapy discontinuation in this study was a lack of efficacy. Interestingly, compared with no use of methotrexate (MTX), concomitant MTX was independently associated with a shorter time to discontinuation of biologics only in first-line therapy, as reported in the PSOLAR study [32,33]. According to this study, the longer treatment duration with biological therapy was in the group of patients with concomitant MTX than in biological agents alone.

### 4.7. Switching of Biologics

In the case of ineffective treatment or adverse effects, the biological agent may be switched to another. According to guidance based on the consensus of experts from 19 European countries, if a PASI 75 response rate is not achieved within 2 months, treatment modification may be required. As reported by the United States review article in 2015, patients who experienced treatment failure to previous anti-TNF-α therapy may have a greater response to another anti-TNF-α drug. Moreover, switching to another agent may be more effective in patients with concomitant psoriatic arthritis, as clinical trials showed the better psoriatic arthritis response rates with adalimumab, etanercept, and infliximab than with ustekinumab [16]. A Japanese study in 2017 has also reported that switching to the second biological therapy usually causes a significant improvement in moderate-to-severe psoriasis. Overall, 275 patients were enrolled in the mentioned study and consisted of the subgroups treated with infliximab, adalimumab, and ustekinumab. A total of 51 patients were required to switch to the second biologic. The main reasons for switching were inefficacy and adverse events. Most inefficacy cases were observed with adalimumab, followed by infliximab and ustekinumab, while the most adverse events appeared with infliximab [1]. In our study, for 15.2% (*n* = 12) of patients, the biological agent was switched to another due to inefficacy; no patients were required to switch to another agent due to an adverse event. In Lithuania, psoriasis treatment is considered ineffective when there is no improvement in the disease (a 75% PASI improvement from baseline) within 3 months of treatment. Additionally, topical treatment and (or) MTX may be added. If there is no remission or disease activity reduction after 6 months of treatment modification, the combination therapy is intolerable or contraindicative, a biological agent may be switched. The first-line biological agents in Lithuania are TNF-α inhibitors, taking into account the lowest annual cost of the treatment. MTX is always given concomitantly with infliximab; therefore, if a patient does not tolerate it or there is a contraindication for MTX use, another TNF-α or IL inhibitor should be given [6].

### 4.8. Comorbidities

Patients with psoriasis often have other comorbidities such as dyslipidemia, type 2 diabetes, obesity or metabolic syndrome, though one of the most common diseases is arterial hypertension [25,34,35,36,37]. In this study, arterial hypertension was diagnosed for 40.5% of all subjects with severe psoriasis. Both hypertension and psoriasis are caused by a chronic inflammatory state. The association between these conditions may be due to several shared pathological pathways, such as overexpression of endothelin in vascular endothelial cells and keratinocytes, increased oxidative stress and inflammatory mechanisms driven by TNF-α and IL-17 [35,36]. Compared with other psoriasis treatments, biological therapy is associated with a lower risk of cardiovascular events [38,39]. However, the US study in 2014 with TNF-α [40] and IL-12/23 inhibitors showed minimal or no reduction in cardiovascular inflammation [25,41].

Because of adipogenesis and lipid metabolism disorders, due to increased inflammatory cytokines, patients with psoriasis are more commonly diagnosed with obesity and dyslipidemia [35,37]. According to our study results, 13.9% (*n* = 11) of all subjects were diagnosed with dyslipidemia and 12.7% (*n* = 10) with obesity.

Very, often the choice of appropriate biological therapy for a patient is determined by the presence of comorbidities. For instance, TNF-α inhibitors, except for etanercept, are considered first-line therapies for moderate to severe Crohn’s disease [42]. Moreover, patients with LTBI can safely use IL-17 inhibitors [43]. Similarly, the ideal drugs to treat psoriasis in obese patients are infliximab and ustekinumab, as they are dosed based on weight [42].

### 4.9. Additional Psoriasis Treatment with Topical Medications, Phototherapy, and Methotrexate

The current data show that, in most cases, combination therapy is more effective and is well tolerated. Moreover, additional topical treatment may sustain the initial effect of biologics, reduce the adverse effects and dosage of both agents and lead to a greater response to biological therapy [44,45,46]. According to the analysis by the International Psoriasis Council in 2016, better treatment efficacy was achieved for patients who were treated with adalimumab in combination with calcipotriol and betamethasone for 1 month. However, after this point of time, a better response rate was observed in the adalimumab monotherapy group [46]. The REFINE study in 2015 compared the health-related quality of life (HRQoL) index for patients who received etanercept 50 mg twice a week versus those who used it with topical corticosteroids; the results for HRQoL were the same in both groups [47]. In our study, all the patients used additional topical treatments and the most frequent topical agents were corticosteroids (84.8%). These results are almost similar to another study in 2017, where 86.0% of patients who received TNF-α inhibitors and 76.7% of those who received ustekinumab also used topical corticosteroids [48].

Phototherapy combined with biologics may also increase the efficacy of psoriasis management. The US review in 2014 has shown the positive effects of etanercept, adalimumab and ustekinumab in combination with narrow-band ultraviolet B (NB-UVB) [45]. In total, 46.8% (*n* = 20) of subjects in our study received NB-UVB phototherapy.

In total, 65.8% of our patients received methotrexate together with biological therapy. The US study in 2014 observed that the PASI 75 response rate was significantly higher in patients with etanercept/MTX combination therapy than in the etanercept monotherapy group (77.3 versus 60.3% respectively; *p* < 0.0001) [45]. Almost the same results, with better PASI 75 and PASI 90 response rates, were published in a German systematic review and metanalysis in 2015 [14]. Furthermore, the Israel trial in 2018 showed that, following treatment failure of etanercept monotherapy in patients with moderate-to-severe psoriasis, etanercept drug survival may be increased by adding a low dose (up to 15 mg) of MTX [12]. Despite the fact that methotrexate and biological combination therapy shows better treatment efficacy, methotrexate is associated with elevated hepatic transaminase levels. It also may cause liver fibrosis and cirrhosis, although these changes are more frequent in those patients who have other risk factors, such as alcohol use, obesity or diabetes [49]. Furthermore, it is noted that improper use of methotrexate may cause other adverse events such as nausea, fever, pancytopenia, mucositis, renal failure, gastrointestinal toxicity or alopecia [50]. In our study, 25% of patients discontinued methotrexate due to intolerance. According to US metanalysis results in 2014, between 15% and 50% of patients who received a long-term low to moderate dose of methotrexate, serum AST and/or ALT values were elevated; however, in most cases, these changes were self-limiting and mild [49]. In our study, 31.3% of patients discontinued methotrexate due to elevated hepatic enzymes. However, according to recommendations from a European consensus meeting, methotrexate (5–15 mg/week) in combination with TNF-α antagonists is safe and it increases the long-term efficacy of the treatment regimen [41]. Moreover, the UK analysis in 2016 showed that methotrexate may reduce the risk of myocardial infarction and cardiovascular diseases in patients with moderate to severe psoriasis [51].

### 4.10. Comorbidities Treatment and Consultations of other Specialists

In this study, 38% of all subjects received antihypertensive treatment, 30.4% hepatoprotectors and 16.5% nonsteroidal anti-inflammatory drugs (NSAIDs). Overall, 30.4% of patients were consulted by cardiologists and 8.9% by rheumatologists. However, we were not able to compare these results because there are no scientific publications about comorbidity treatments and consultations of other specialists for patients with psoriasis.

### 4.11. Future Perspectives

Currently, there are some novel biological agents that offer hopesfor more efficient psoriasis treatment [2]. Two of them are IL-17A inhibitors: ixekizumab and secukinumab. A metanalysis in 2017 showed higher PASI 75, PASI 90 and PASI 100 response rates for psoriasis patients receiving IL-17A inhibitors compared with a placebo group. However, the adverse events were more common for IL-17A inhibitors than in the placebo group, but there was no significant difference in serious adverse events between those groups [7]. Moreover, a Cohrane metanalysis in 2017 concluded that ixekizumab and secukinumab were significantly more effective than TNF-α inhibitors infliximab, adalimumab and etanercept [17].

Other novel agents are the IL-23 inhibitors guselkumab and risankizumab. In 2017, a phase III multicenter, randomized VOYAGE 2 study showed higher PASI 75 and PASI 100 response rates in patients receiving guselkumab in comparison to those who were receiving the TNF-α inhibitor adalimumab [2]. Furthermore, the results of the UK study in 2019 showed that risankizumab had a higher efficacy than ustekinumab and a placebo for moderate to severe psoriasis. Moreover, adverse event profiles were similar among these groups [52].

### 4.12. Limitations

There are some limitations to our study. Firstly, the sample size in this study was small. Secondly, very few patients were receiving long-term biological therapy. Moreover, adalimumab was not available to adult patients until 2019, when it became the first line biologic in Lithuania; therefore, a comparison of efficacy with infliximab and etanercept is lacking. To strengthen the results, this study and their significance, a larger sample size with a longer treatment period is required.

## 5. Conclusions

The majority of patients on biologics were male, and the most commonly used biological agent was infliximab. Infliximab had the highest efficacy when reaching PASI 90 and the incidence rate of adverse effects in this biological agent were lower than in others. Ustekinumab had the highest proportion of patients reaching PASI 50 and 75 and it had the best drug survival rates. The mean BMI of all subjects slightly increased. Changes in hematology and biochemistry laboratory tests were insignificant. Although some patients tested positive for Mantoux and QuantiFERON-TB Gold tests, none of their X-rays showed tuberculosis-specific changes. The most common comorbidity was arterial hypertension and more than one third of the subjects were taking antihypertensive drugs. Additional treatment with topical medications has been prescribed for all patients, and more than half of them have been taking methotrexate together with biological therapy.

## Figures and Tables

**Figure 1 medicina-56-00275-f001:**
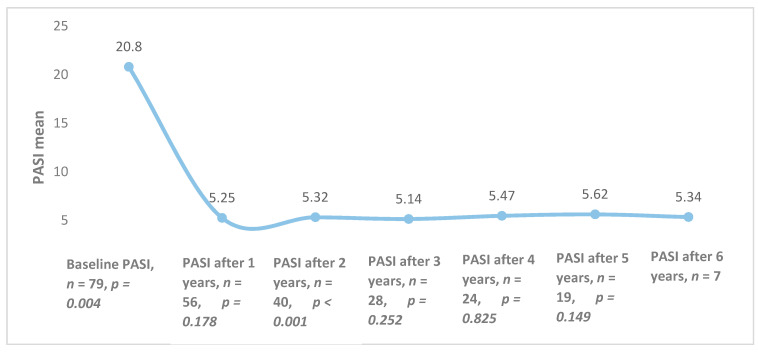
Combined Psoriasis Area and Severity Index (PASI) averages of all patients during treatment with biological therapy.

**Figure 2 medicina-56-00275-f002:**
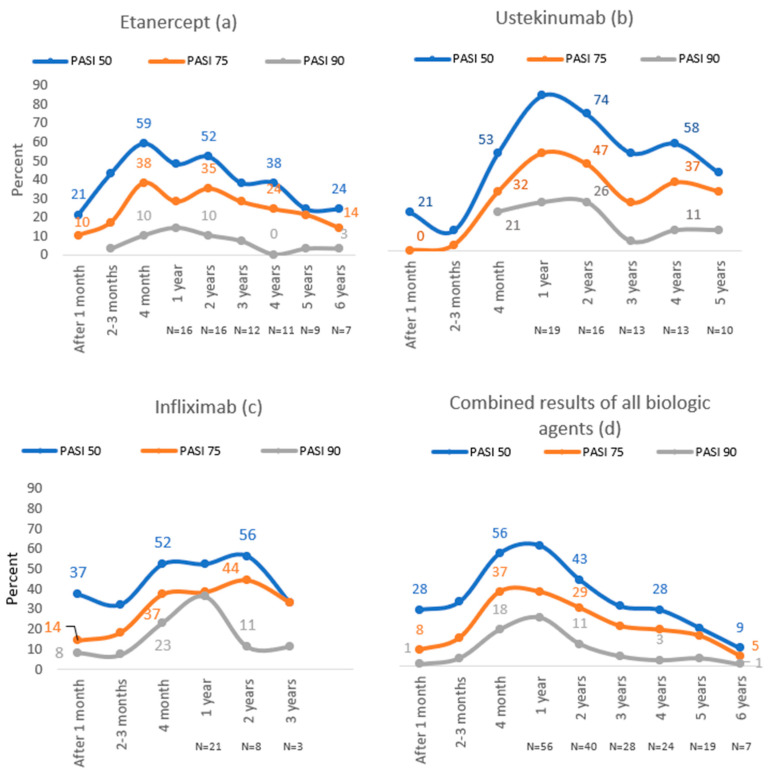
PASI 50, PASI 75 and PASI 90 results throughout 6 years of follow-up with etanercept (**a**), ustekinumab (**b**), infliximab (**c**) and combined results of all biologic agents (**d**).

**Figure 3 medicina-56-00275-f003:**
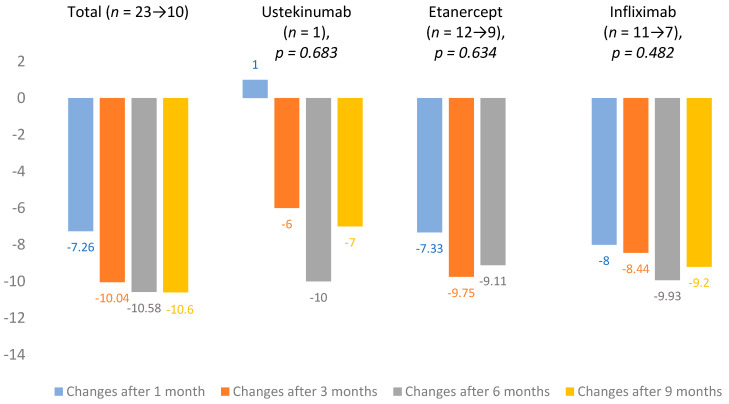
Dermatological quality of life index (DLQI) mean changes.

**Figure 4 medicina-56-00275-f004:**
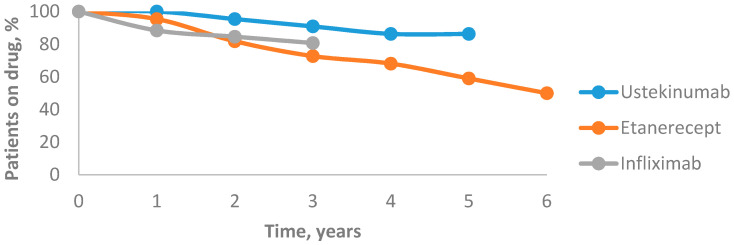
Drug survival rates of biological agents.

**Table 1 medicina-56-00275-t001:** Baseline demographic and clinical characteristics.

Baseline Demographic and Clinical Characteristics	Ustekinumab	Etanercept	Infliximab	*p* Value	Total
**Number of patients treated by the first biologic** **, *n* (%)**	19 (24.1)	29 (36.7)	31 (39.2)	0.900	79 (100)
**Males, *n (%)***	14 (73.7)	21 (72.4)	24 (77.4)		59 (74.7)
**Females, *n (%)***	5 (26.3)	8 (27.6)	7 (22.6)		20 (25.3)
**Age at enrolment, mean, years (SD)**	49.5 (10.6)	48.1 (12.3)	45.5 (11.1)	0.461	47.4 (11.3)
**Family history of psoriasis, *n* (%),**					
**Positive** **Negative** **No data**	5 (26.3) 4 (21.1) 10 (52.6)	7 (24.1) 3 (10.3) 19 (65.5)	14 (45.2) 8 (25.8) 9 (29.0)	0.74	26 (100) 15 (100) 38 (100)
**Current smokers, *n* (%)**	1 (5.3)	4 (13.8)	5 (16.1)	0.519	10 (12.7)
**Baseline BMI, kg/m^2^ (SD)**	27.8 (4.2)	26.5 (5.1)	28.7 (7.4)	0.36	27.7 (5.6)
**Mean morbidity at baseline, years (SD)**	33.5 (13.9)	24.4 (11.6)	22.1 (10.5)	0.005	25.7 (13)
**Duration of first biological treatment, mean, weeks (SD)**	219.1 (95.7)	156.2 (137.4)	85.9 (50.2)	<0.001	143.7 (112.4)
**Second biologic, *n* (%)**	2 (25)	3 (37)	3 (38)	-	8 (100)
**Third biologic, *n* (%)**	2 (100)	-	-	-	2 (100)
**Psoriatic arthropathy, *n* (%)**	11 (57.9)	20 (69)	26 (83.9)	0.123	57 (72.2)
**Psoriatic onychodystrophy, *n* (%)**	17 (89.5)	23 (79.3)	24 (77.4)	0.549	64 (81)

Standard deviation (SD); body mass index (BMI); Psoriasis Area and Severity Index (PASI).

**Table 2 medicina-56-00275-t002:** Changes in complete blood count, biochemical blood parameters, average values of liver function tests.

Laboratory Parameters	Baseline Count, Mean (SD)	After 1 Year, Mean (SD)	After 2 Years, Mean (SD)	After 3 Years, Mean (SD)	After 4 Years, Mean (SD)	After 5 Years, Mean (SD)	After 6 Years, Mean (SD)
**WBC (×10^9^/L)**	6.7 (2.2)	6.1 (1.4)	6.6 (3.2)	6.3 (1.5)	6.2 (1.4)	6.2 (1.2)	6.3 (0.9)
**RBC (×10^12^/L)**	4.8 (0.6)	4.9 (0.5)	4.9 (0.4)	5.0 (0.4)	4.9 (0.4)	5.0 (0.4)	4.7 (0.5)
**Hemoglobin (g/L)**	144.8 (16)	147.8 (13.8)	149.9 (12.3)	151.1 (14.5)	149.2 (13.5)	149.9 (12.6)	143.1 (16.6)
**Platelet count (×10^9^/L)**	235.7 (59.3)	222.5 (42.5)	223.4 (40.6)	219.5 (36.9)	221.6 (40.4)	217.3 (34.2)	261.9 (62.3)
**ESR (mm/h)**	12.8 (14.8)	9.5 (9.5)	9 (10.3)	7.5 (8.5)	9.9 (10.2)	12 (12.5)	13.6 (18.9)
**CRP (mmol/L)**	7 (15.04)	2.4 (3.4)	6.5 (27.6)	1.8 (2.4)	1.6 (2.4)	2.3 (4.6)	1.7 (1.9)
**Glucose (mmol/L)**	5.3 (1.1)	5.3 (0.7)	5.3 (0.8)	5.5 (0.9)	5.4 (0.78)	5.7 (1)	6.1 (1.2)
**Cholesterol (mmol/L)**	5.1 (0.8)	5.2 (0.8)	5.4 (0.8)	5.4 (0.7)	5.4 (0.8)	5.5 (0.8)	5.5 (0.6)
**AST (U/L)**	24.6 (14.7)	28 (17.3)	25 (8.1)	29.7 (25.2)	25.8 (9.4)	25.2 (6.4)	21.8 (5.3)
**ALT (U/L)**	35.4 (42.1)	37.4 (20.1)	35.8 (20.2)	39.2 (25.3)	33.1 (18.1)	31 (14.4)	23.1 (6)
**ALP (U/L)**	78 (23.6)	75.5 (25.6)	78.1 (39.6)	76.7 (27.8)	74.3 (22.8)	74.5 (20.9)	79.7 (20.2)
**GGT (U/L)**	32.3 (37.8)	40.1 (39.3)	44.3 (61.8)	37.8 (25.6)	42 (39.7)	46.3 (56.8)	33.4 (22.3)

Standard deviation (SD); white blood cell (WBC); red blood cell (RBC); erythrocyte sedimentation rate (ESR); C-reactive protein (CRP); gamma-glutamyl transferase (GGT); alkaline phosphatase (ALP); alanine aminotransferase (ALT); aspartate aminotransferase (AST).

**Table 3 medicina-56-00275-t003:** Adverse effects of biological therapy.

Adverse Effects	Ustekinumab, Cases Per Patient Year	Etanercept, Cases Per Patient Year	Infliximab, Cases Per Patient Year
**Upper respiratory tract infections**	0.337	0.436	0.151
**Dizziness**	-	0.011	-
**Nausea**	-	0.011	-
**Mood swings**	-	-	0.042
**Fever**	-	0.011	-
**Latent tuberculosis**	0.037	0.022	-

**Table 4 medicina-56-00275-t004:** Dermatological and non-dermatological comorbidities.

Dermatological Comorbidities, *n (%), p =* 0.062		Non-Dermatological Comorbidities, *n (%), p =* 0.867	
**Verruca vulgaris**	8 (10.1)	**Hypertension**	32 (40.5)
**Seborrheic dermatitis**	8 (10.1)	**Dyslipidemia**	11 (13.9)
**Onychomycosis**	5 (6.3)	**Obesity**	10 (12.7)
**Pityriasis versicolor**	4 (5.1)	**Hypertensive cardiomyopathy**	8 (10.1)
**Melanocytic nevus**	4 (5.1)	**Latent tuberculosis infection**	8 (10.1)
**Acne vulgaris**	3 (3.8)	**Diabetes mellitus, type 2**	4 (5.1)
**Actinic keratosis**	2 (2.5)	**Metabolic syndrome**	3 (3.8)
**Total**	25 (31.6)	**Total**	50 (63.3)

**Table 5 medicina-56-00275-t005:** Additional topical treatment and phototherapy.

Additional Topical Treatment, *n (%)*		Phototherapy, *n (%)*	
**Emollients**	78 (98.7)	**UVB311 phototherapy whole-body**	36 (45.6)
**Topical glucocorticoids**	67 (84.8)	**UVB311 for hands**	10 (12.7)
**Keratolytics**	41 (51.9)	**UVB311 comb**	6 (7.6)
**Topical combination of vitamin D analogue and glucocorticoids**	20 (25.3)	**UVB311 for feet**	5 (6.3)
**Vitamin D analogues**	8 (10.1)	**Combination therapy of UVA and UVB**	3 (3.8)
**Calcineurin inhibitors**	4 (5.1)	**Topical PUVA therapy**	1 (1.3)
**Total**	79 (100)	**Total**	37 (46.8)

**Table 6 medicina-56-00275-t006:** Consultations of other specialists and medications used.

Medications Used, *n (%)*		Other Specialist Consultations, *n* (%)	
**Antihypertensives**	30 (38)	**Cardiologists**	30.4 (24)
**Hepatoprotectors**	24 (30.4)	**Rheumatologists**	8.9 (7)
**Antifungals**	14 (17.7)	**Gastroenterologists**	7.7 (6)
**NSAIDs**	13 (16.5)	**Endocrinologists**	3 (3.8)
**Systemic glucocorticoids**	10 (12.7)	**Gynaecologists**	1 (1.3)
**Topical antibiotics**	8 (10.1)	**Psychiatrists**	1 (1.3)
**Total**	23 (29.1)	**Total**	33 (41.8)

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
