# Peer review of "A Six-Year Analysis of Biological Therapy for Severe Psoriasis in a Lithuanian Reference Centre of Dermatovenereology"

_medicina, 2020, doi:10.3390/medicina56060275_

Round 1

Reviewer 1 Report

This paper describes a real world data of psoriatic patients treated by biologics at a single facility. Though the size is small, it is well documented and discuss all aspects of psoriatic patients and therapy. However, there are some points to be cleared. 

Major points.

Table 1. has 79 total subjects. Does this 79 patients contain duplication? If not, the firs row of the table should be clearly labeled as “Number of patients treated by the first bio”. Also, though described in the text, it is better to show the total in the table, so that readers can easily grab the information of the whole participants.

Author Response

Point 1: Table 1. has 79 total subjects. Does this 79 patients contain duplication? If not, the firs row of the table should be clearly labeled as “Number of patients treated by the first bio”.

Response 1: All these 79 patients do not contain duplication. So we labelled it as “Number of patients treated by the first biologic” in the new version of manuscript.

Point 2: Also, though described in the text, it is better to show the total in the table, so that readers can easily grab the information of the whole participants.

Response 2: As per advice, we have added the totals in the table.  

Reviewer 2 Report

I have one suggestion: In the discussion section under comorbidities (line 372 onwards), please also mention that very often the choice of appropriate biologic therapy for a patient is determined by the presence of comorbidities. For instance, infliximab and ustekinumab are dosed based on weight and are ideal drugs to treat psoriasis in obese patients. Similarly,  Anti-TNF-alpha agents except for etanercept are considered first-line management of moderate to severe Crohn’s Disease.

Please refer to the following articles:

www.ncbi.nlm.nih.gov/pubmed/30017706

www.ncbi.nlm.nih.gov/pubmed/30017705 

Author Response

Point 1: I have one suggestion: In the discussion section under comorbidities (line 372 onwards), please also mention that very often the choice of appropriate biologic therapy for a patient is determined by the presence of comorbidities. For instance, infliximab and ustekinumab are dosed based on weight and are ideal drugs to treat psoriasis in obese patients. Similarly,  Anti-TNF-alpha agents except for etanercept are considered first-line management of moderate to severe Crohn’s Disease.

Please refer to the following articles:

www.ncbi.nlm.nih.gov/pubmed/30017706

www.ncbi.nlm.nih.gov/pubmed/30017705 

Response 1: As per advice, we have added this information and referred it to provided articles.
